# The Influence of Ambient Temperature on High Performance Concrete Properties

**DOI:** 10.3390/ma13204646

**Published:** 2020-10-18

**Authors:** Alina Kaleta-Jurowska, Krystian Jurowski

**Affiliations:** Faculty of Civil Engineering and Architecture, Opole University of Technology, Katowicka 48, 45-061 Opole, Poland; k.jurowski@po.edu.pl

**Keywords:** concrete, temperature, high performance concrete (HPC)

## Abstract

This paper presents the results of tests on high performance concrete (HPC) prepared and cured at various ambient temperatures, ranging from 12 °C to 30 °C (the compressive strength and concrete mix density were also tested at 40 °C). Special attention was paid to maintaining the assumed temperature of the mixture components during its preparation and maintaining the assumed curing temperature. The properties of a fresh concrete mixture (consistency, air content, density) and properties of hardened concrete (density, water absorption, depth of water penetration under pressure, compressive strength, and freeze–thaw durability of hardened concrete) were studied. It has been shown that increased temperature (30 °C) has a significant effect on loss of workability. The studies used the concrete slump test, the flow table test, and the Vebe test. A decrease in the slump and flow diameter and an increase in the Vebe time were observed. It has been shown that an increase in concrete curing temperature causes an increase in early compressive strength. After 3 days of curing, compared with concrete curing at 20 °C, an 18% increase in compressive strength was observed at 40 °C, while concrete curing at 12 °C had a compressive strength which was 11% lower. An increase in temperature lowers the compressive strength after a period longer than 28 days. After two years of curing, concrete curing at 12 °C achieved a compressive strength 13% higher than that of concrete curing at 40 °C. Freeze–thaw performance tests of HPC in the presence of NaCl demonstrated that this concrete showed high freeze–thaw resistance and de-icing materials (surface scaling of this concrete is minimal) regardless of the temperature of the curing process, from 12 °C to 30 °C.

## 1. Introduction

### 1.1. Temperature Influence on the Hydration Process of Portland Cement

Temperature is an important factor which influences the hydration process of cement and the properties of concrete mixture and hardened concrete. It is known that the rate of reaction of cement hydration grows with increasing temperature. The consequence of this is a faster increase in the strength of concrete in the early stage of maturation [1,2,3].

The influence of temperature on the cement hydration process has been the subject of many studies. It has been found that, in the early stages of maturation, the rate of hydration of the alite significantly increases along with an increase in temperature, but later (from 28 to 90 days) it decreases depending on the type of cement [4]. After a year of maturation, the highest degree of hydration was observed in cement pastes cured at 10 °C and the lowest in those at 60 °C. Furthermore, it was found that, in cement paste curing at 10 °C, almost all the cement grains were hydrated, while at 60 °C cement grains which were only partially hydrated could be observed.

Studies on the microstructure of hydrated cement phases at different temperatures have shown that temperature also influences the morphology, type, and number of hydrate phases formed. At higher temperatures, the more heterogeneous distribution of hydrate phases and formations of shorter needle-shaped ettringite crystals are observed [5]. Moreover, the results indicate that at elevated temperatures the hydration rate of alite and belite is higher.

The results of the authors of the study [6] indicate that the apparent density of cement paste increases with temperature (in the range from 5 °C to 60 °C). According to the authors, this is due to the reduction of bonded water. This results in a more porous microstructure of the cement paste and a reduction in the volume occupied by the C-S-H phase. Higher porosity of cement pastes cured at elevated temperatures was also found by the authors [7]. The result is lower strength of the paste and lower durability of the resulting material.

The studies presented in the paper [8] indicate that at elevated temperatures (40 °C and 50 °C) the formation of the C-S-H phase with higher density, more heterogeneous distribution of hydration products and higher porosity were observed. At 50 °C, calcium monosulphate was observed in the initial period, while the amount of ettringite significantly decreased. This was also confirmed by the authors of other works [9,10]. Due to the increase in porosity, the strength decreases later on. An increase in the porosity of cement pastes cured at elevated temperatures was also observed in binders containing granulated blast-furnace slag [11,12,13,14].

Cement pastes with the addition of fly ash, volcanic ash or granulated blast-furnace slag, cured in the temperature range from 10 °C to 60 °C were tested in the study [11]. It was found that blast furnace slag was the only additive that positively influenced the strength (in relation to the strength of the cement paste without additives), especially at 60 °C. According to these authors, the microstructure of cement pastes cured at 60 °C showed higher porosity than the microstructure of grouts cured at 10 °C.

In [15], it was found that the microstructure of cement paste with silica fume, cured at 23 °C, is homogeneous. This cement paste has a much less porous structure compared to a cement paste without an additive, with the same degree of cement hydration. On the other hand, cement pastes cured at 30 °C and 70 °C differ from cement pastes cured at 23 °C with their $\mathrm{Ca}{\left(\mathrm{OH}\right)}_{2}$ concentration. While the distribution of hydration products is still relatively homogenous, there are larger continuous pores between the cement grains. The authors found that the temperature of curing has a greater impact on the microstructure of a cement paste with silica fume than cement paste without this additive.

Tests of cement pastes cured at temperatures ranging from 5 °C to 50 °C carried out after a longer period of time (up to 91 days) showed that the cement pastes cured at the lowest temperature was hydrated to the greatest extent [16]. These authors have also shown that at a higher curing temperature in a cement paste, the distribution of hydration products is uneven, resulting in a lower compressive strength of these cement pastes after a longer curing time [17,18].

Summing up the results of the research conducted by various authors, it should be stated that the increase in temperature leads to the acceleration of the hydration process of Portland cement, with the distribution of hydration products being more irregular. This results in increased compressive strength in the early stages of curing. Increased temperature also makes the distribution of cement hydration products uneven and increases the porosity of the resulting structure. The consequence of this is a reduction in compressive strength after a longer curing period. This also applies to cement pastes containing mineral additives, although, in the case of additives, such as fly ash or granulated blast-furnace slag, the scale of the phenomenon is smaller, which can be explained by the reduction in hydration heat of binders with these mineral additives.

### 1.2. Influence of Temperature on the Properties of a Fresh Concrete Mixture and Hardened Concrete

The influence of temperature on the cement hydration is reflected in the properties of the concrete mixture and hardened concrete. The production of concrete mixtures at elevated temperatures causes many problems due to the accelerated hydration process of the cement. In addition, the concrete mix has a higher water demand due to evaporation. The influence of temperature on the workability of normal strength concrete is well recognised—increasing temperature leads to workability deterioration [19,20]. The authors of the paper [21] also stated that there is an optimal temperature (about 20 °C) allowing them to obtain a concrete mixture with the most advantageous workability. Klieger [22] found that with the temperature increase of 11 °C, the slump decreases by 25 mm, the result of which it is necessary to increase the water content to maintain its consistency.

The consistency of the concrete mixture also depends on the effectiveness of chemical admixtures at elevated temperatures. Schmidt et al. [23] demonstrated that the behaviour of Self-Compacting Concrete (SCC), containing a superplasticizer, at different temperatures, is different from that of normal concrete. Superplasticizers in a concrete mixture, depending on their chemical structure, have different effects on rheological properties of the concrete mixture. A linear relationship between the temperature and the yield stress of the concrete mixture was shown. The higher the temperature, the faster the yield stress increases [24].

The paper in [25] shows that the temperature of concrete mixture also has an influen on the initian and final setting time of cement. The difference between the initial and the final setting time of the cement decreases as the ambient temperature increases. Moreover, the study [26] shows that an increase in the cement content results in an increase in the temperature of the concrete mixture, as well as a shortening of the setting time.

An increase in ambient temperature generally results in a loss of workability of the concrete mixture. The reason for this phenomenon is both the acceleration of the cement setting process and the faster evaporation of the mixing water at higher temperatures.

The influence of temperature on the properties of hardened concrete is similar to that of cement pastes [27]. Increasing the temperature of concrete curing results in higher early concrete strength; however, the strength decreases after as time goes on. An increase in temperature also reduces the corrosion resistance of concrete [5,28]. This effect is most evident when the concrete mixture is exposed to high temperatures immediately after casting.

The most susceptible to excessive heating are massive elements, where the cooling surface is small in relation to the mass of the concrete mixture being caste. The negative phenomena caused by excessive heating can be minimized by proper selection of binder composition [29].

There are methods to minimise the adverse effects of increased temperature on concrete properties. These include: reducing the cement content in concrete; partial replacement of the cement by mineral pozzolanic and hydraulic additives; the use of cement with low hydration heat; thermal control of aggregates; the use of cool water or the addition of crushed ice to the concrete mixture. In practice, good results are achieved by introducing granulated blast-furnace slag and fly ash into the cement [30,31,32].

The influence of temperature on the properties of normal strength concrete is widely recognised. An increase in the curing temperature also increases the early strength; however, it reduces the strength of the concrete later on and has a negative impact on its durability, which is related to the cement hydration process. However, it should be noted that the influence of temperature on the properties of High Performance Concrete (HPC), which is particularly sensitive to temperature changes due to the relatively low w/c ratio and the use of high range water reducing admixtures (HRWR), is much less well-known.

Apart from the heat generated by the hydration reaction, the temperature of the concrete mixture is also influenced by the temperature of the mixture components, the ambient temperature, and the heat generated by friction as a result of mixing. The temperature of the aggregate is of particular importance, because its content in concrete is relatively high. The temperature of aggregate and water generally corresponds to the ambient temperature, while the temperature of cement stored in silos can be much higher, which further increases the temperature of the concrete mixture.

This paper presents the results of a study on the influence of temperature on the properties of a fresh concrete mixture and hardened HPC containing a polycarboxylate superplasticizer and an addition of silica fume. The tests were carried out at both increased (30 °C) and lowered (12 °C) concrete curing temperatures, but within the range of the practical applicability of concrete. The compressive strength and concrete mix density were tested at temperatures of 12 °C, 20 °C, 30 °C and 40 °C. Special attention was paid to achieving the desired temperature of the mixture components and maintaining this temperature while making the mixture and curing the concrete.

## 2. Materials

The concrete (HPC) was made with Portland cement CEM I 42.5 R (CEM I), with a specific surface area (Blaine) of 440 m^2^/kg. The chemical composition of the cement is shown in Table 1. The results of particle size distribution tests performed with a laser grain analyser are presented in Figure 1.

A superplasticizer based on polycarboxylates (SP) was used as a HRWR. The SP was added in the amount of 1.5% in relation to the mass of cement.

Silica fume (SF) was used as the mineral additive, an amount of 10% in relation to the mass of cement. According to the manufacturer’s specification, the chemical composition of SF is as follows: $\mathrm{Si}O_{2}$ (min. 85%), $Fe_{2}O_{3}$ (max. 2.5%), CaO (max. 1.0%) and $Al_{2}O_{3}$ (max. 1.5%).

The concrete mixture was made of natural fine aggregate (0/2 mm fraction) and crushed basalt aggregate (fractions 2/8 and 8/16 mm). The particle size distribution of individual aggregate fractions is shown in Figure 2.

The results of the physical properties of aggregates, such as bulk density, specific density and water absorption are presented in Table 2.

The particle size distribution was selected using the iterative method described by Kuczyński [33,34]. When composing the particle size distribution from several different aggregate fractions, they were put together in such a way as to ensure the greatest possible tightness, with the lowest possible water demand.

The composition of the HPC mixture was designed using the experimental method, assuming the particle size distribution, as defined above, and the amount and type of cement with silica fume added (Table 3). The w/c ratio was selected to obtain concrete with a compressive strength of more than 100 MPa. The designed concrete mixture composition is presented in Table 3. The consistency of the concrete mixture was regulated by adding an appropriate amount of the SP.

The ingredients were mixed in a forced circulation mixer by Zyklos Mixer ZK 150 HE (Pemat, Freisbach, Germany). The same procedure for adding ingredients to the mixer and a constant mixing time of the concrete mixture at all temperatures were applied. The mixing procedure used is shown in Table 4.

Concrete mixtures were made at temperatures of 12 °C, 20 °C, 30 °C and 40 °C. In order to stabilise the temperature of the components, the cement, silica fume, aggregate and water were kept at a controlled, assumed temperature for at least 72 h before the concrete mix was made, using climate chambers.

Every effort was made to keep the temperature in the room where the concrete mix was prepared at the assumed level. The temperature was increased using an appropriate heating system and air heater units. Tests at lowered temperatures were carried out in the winter period, which made it possible to maintain the assumed temperature.

## 3. Methods

### 3.1. Tests of the Properties of the Materials Used

The particle size distribution of cement was tested with the use of the Mastersizer 3000 laser particle analyser (Malvern, UK), and the wet method. Isopropanol was used as the dispersant. The test was carried out with an obscuration range of 10–15%. The presented test results are an average of at least 3 measurements.

Physical properties of aggregates such as bulk density, specific density and water absorption were tested in accordance with standards EN 1097-3:2000, EN 1097-6:2013 and EN 1097-6:2013, respectively.

### 3.2. Concrete Mixture Testing

Consistency testing by the concrete slump test was performed according to EN 12350-2:2011. This test consists of placing and compacting the concrete mixture in a truncated cone shaped mould. The result of the test is the decrease in the height of the concrete mixture immediately after removing the mould.

Consistency determination with the flow table test was carried out according to EN 12350-5:2011. This test consists of placing the mixture in a truncated cone on a plate and then, after lifting the mould, 15 cycles of lifting and free falling of the upper table plate are performed. The result of the test is the flow diameter of concrete mixture.

Consistency determination with the Vebe test was carried out according to EN 12350-3:2011. The result of the test is the time of vibration of the mixture placed in the cylinder until the level of the mixture in the cylinder is completely consolidated.

Testing of the consistency of the concrete mixture using the concrete slump test, the flow table test and the Vebe test was carried out immediately after mixing the ingredients, as well as after 30 and 60 min. The concrete mixture was mixed at slow speed until testing.

The air content of the concrete mixture was determined in accordance with EN 12350-7:2011 by means of a pressure gauge. Testing using a pressure gauge is based on Boyle–Mariotte’s law and consists in the fact that a known volume of air at a certain pressure combines in a tightly closed container with an unknown volume of air contained in a concrete mixture sample.

The density of the concrete mixture was determined according to EN 12350-6:2011. This method consists in determining the weight of a concrete mixture, completely filling a container with a known volume.

### 3.3. Testing on Hardened Concrete

The density of the concrete was determined according to EN 12390-7:2011, hydrostatic method. The method consists in determining the mass and volume of a concrete sample by determining the mass of displaced water. The test was carried out on cubic samples of 100 mm side.

Tests of concrete water absorption were carried out using the method described in the PN-B-06250:1988 standard, after 28 days of concrete curing. Each time the test was performed on 3 cubic samples with a side dimension of 100 mm. The samples were first saturated in water to constant mass and then dried at 105 °C to constant mass.

The compressive strength was tested according to EN 12390-3:2011. The test was carried out using the Controls MCC8 strength press (Controls Group, Liscate, Italy). The test was carried out on cubic samples of 100 mm side. The concrete samples cured for more than 28 days were taken out of water and cured at laboratory temperature (20 ± 2) °C until the test.

Testing of the depth of penetration of water under pressure of the concrete was carried out in accordance with EN 12390-8:2011. Each time, the test was carried out on 6 cubic samples of concrete with a side of 150 mm, cured before the test for 28 days in water.

The tests of concrete resistance to cyclic freezing and thawing in the environment of de-icing salt were performed according to EN 12390-9:2007, using the ”slab test”. This test consists in determining the weight of scaled material from the concrete sample surface after 7, 14, 28, 42 and 56 freezing and thawing cycles in the presence of 3% NaCl solution. Then, 4 cubic samples of concrete with a side of 150 mm were used for the test, which were cured for 7 days at 12 °C, 20 °C or 30 °C. The remaining curing and preparation period of the test was carried out in accordance with EN 12390-9:2007.

The freeze–thaw durability was also tested according to Polish standard PN-B-06250:1988. The test is carried out on 12 cubic concrete samples with a side dimension of 100 mm, 6 of which undergo 300 freeze/thaw cycles. The result of the test is a loss of compressive strength of the tested samples, compared to the other 6 ”witness samples’’. Before testing, the samples were cured for 28 days at 12 °C, 20 °C or 30 °C water temperatures.

All test results presented in this paper are average values for a minimum of three measurements. The uncertainty values given are the expanded uncertainty of measurement with an expansion probability of approximately 95% and a corresponding expansion factor of *k* = 4.30 (for 3 samples) *k* = 3.18 (for 4 samples) and *k* = 2.57 (for 6 samples).

## 4. Results and Discussion

### 4.1. Temperature Influence on the Properties of the Concrete Mixture

Consistencies in the concrete mixture were tested by three methods. This was necessary because consistency differed significantly from one temperature to another and only one of the methods give results outside the range of applicability of this method.

Figure 3 shows the slump depending on the time of the test and the ambient temperature at which the concrete mixture was prepared from ingredients that had previously obtained the same temperature as the ambient temperature. It is clear that the concrete mixture preparation temperature in the range from 12 °C to 30 °C has a significant impact on the consistency. At 30 °C, immediately after mixing the ingredients, a minimum slump (20 mm) is observed, while at 12 °C this value is as high as 280 mm, which is beyond the applicability range of this test method.

Similar results were obtained using the flow table test and the Vebe test. The flow diameters and Vebe time depending on the temperature as well as the time of the test are shown in Figure 4 and Figure 5, respectively.

The results of the consistency test using the flow table test (Figure 4) showed a decrease in the flow diameter of the concrete mixture as the temperature and time increase. The Vebe time needed for compaction of the concrete mixture also increased with increasing temperature (Figure 5).

The workability of the concrete mixture has also decreased significantly in up to 60 min. Moreover, at 30 °C, the decrease in flowability was so great that after only 30 min correct measurement was impossible (both the Vebe test and the flow table test).

The presented results of the concrete mixture consistency tests showed that an increase in temperature causes a loss of workability. This effect is particularly visible at 30 °C. At this temperature, the workability of the concrete mixture is lost after only 30 min. The observed phenomenon is mainly the effect of increasing the rate of water evaporation from the concrete mixture at elevated temperature and accelerated cement hydration process [19,21], as well as from low ratio w/b=0.27 in HPC. The results of consistency tests over time are difficult to assess as no similar tests of an HPC concrete mixture were found in the literature. The loss of workability can also be caused by a decrease in the effectiveness of the HRWR admixture at elevated temperatures [23,35]. In practice, it would be necessary to use more of the HRWR admixture. In this paper, the proportions of ingredients have been retained to ensure the comparability of test results.

The results of air content tests in the concrete mixture depending on the curing temperature and testing time are presented in Figure 6.

As shown in Figure 6, the air content in the concrete mixture generally increases with the mixture temperature and does not change significantly over time. It should be noted that the results obtained at 30 °C may be unreliable as the consolidation of concrete mix was difficult at this temperature.

The concrete mixture density tests showed that the temperature in the tested range (from 12 °C to 40 °C) does not have a significant influence on this parameter. Differences in the results of density determination did not exceed the measurement error (Table 5). Similar results were obtained by the authors in the paper [36], who demonstrated that the differences in the density of the concrete mixture tested, at different temperatures, are negligible.

### 4.2. Temperature Influence on the Properties of Hardened Concrete

The concrete density tests showed that in the range from 12 °C to 30 °C, an increase in temperature causes a slight increase in the concrete density. However, it should be noted that these differences are minimal, slightly exceeding the error of measurement of the test method (Table 6).

The results of the water absorption tests of the samples of concrete cured at different temperatures showed that in all cases the obtained values were at a very low level. The least absorbent is the concrete cured at 20 °C, and the largest is the concrete cured at 30 °C (Figure 7), however the presented differences should be considered as small. In addition, at elevated temperatures the water absorption rate increases more than at the lower curing temperature (12 °C). Similar results were obtained by the authors of the paper [37] in testing mortars.

The results of compressive strength testing of concrete samples depending on curing temperature (12 °C, 20 °C, 30 °C and 40 °C) are shown in Table 7. The increase in compressive strength over time of these concrete is shown in Figure 8.

As can be seen from the data in Figure 8, the early strength (after 3 and 7 days) of concrete samples cured at elevated temperatures (30 °C and 40 °C) is higher than that of concrete samples cured at lower temperatures (12 °C and 20 °C). However, after a longer curing period (from 90 days to 2 years), the highest strength was demonstrated for concrete cured at lower temperatures. The data presented in Figure 8 show that for concrete curing at almost all tested temperatures, the strength is similar at age of about 40 days.

The increase in the compressive strength of concrete in the initial period of time (up to 7 days), with an increase in the curing temperature, can be explained by the acceleration of the cement hydration process [38]. However, as the results of the research [25] show, an increase in the rate of cement hydration contributes to the formation of a weaker, more porous microstructure, which in effect results in lower compressive strength after a longer curing period (over 28 days).

Concrete cured at 12 °C and 20 °C achieved similar strength values at the age of 2 years (about 130 MPa), while concrete made at 20 °C achieved this strength much faster. It can therefore be concluded that a temperature of 20 °C is the most favourable, due to the increase in strength over time. Decreased early concrete strength at low temperatures is also confirmed by the results of the research presented, for example, in [22].

The results of penetration depth tests of the concrete samples cured at 12 °C, 20 °C and 30 °C showed that no significant differences were observed in the values of this parameter (Figure 9) which is in the 9–11 mm range. The above shows a very high tightness of the tested concrete and indicates a properly selected composition of the concrete mixture, including a tight particle size distribution.

The high concrete tightness shown in the test for the depth of penetration of water under pressure contributes to the durability of the concrete. This is confirmed by the results of the freeze–thaw durability tests carried out with two methods: the method according to PN-B-06250 and the method according to EN-12390-9.

The results of concrete freeze–thaw durability tests using the method according to PN-B-06250 are presented in Table 8. The tests have shown that, regardless of the concrete curing temperature (12 °C, 20 °C and 30 °C), this concrete can be classified as having the highest degree of freeze–thaw durability (F300 according to PN-B-06250).

The HPC subjected to 300 freezing and thawing cycles shows no cracking or significant mass loss. The decrease in compressive strength is highest for the concrete prepared at 12 °C (8.6%) and lowest for the concrete prepared at 30 °C (1.7%). The reason for the observed phenomenon may be the fact that, on the day of the beginning of freezing (28th day of maturation), the concrete samples curing at lower temperatures had not yet obtained sufficiently high strength.

The results of tests of resistance of the concrete samples matured at 12, 20 and 30 °C to cyclic freezing and thawing in the presence of de-icing salt are shown in the table (Table 9).

The scaling masses under the influence of cyclical freezing and thawing of concrete with simultaneous action of salt solution were minimal. Thus, the influence of the concrete curing temperature on the freeze–thaw durability result of this method has not been demonstrated. At the same time, the assessment according to the Borås criterion proves that the designed concrete is of very good quality, regardless of the curing temperature applied.

## 5. Conclusions

HPC is increasingly used in civil engineering. The assessment of the influence of temperature on the properties of concrete is particularly important because of the application of HPC concrete at both elevated and lowered temperatures. This paper presents the results of research on the influence of ambient and curing temperature on the properties of the concrete mixture and hardened HPC.

Consistency tests carried out using various methods have shown that HPC mixtures made at different ambient temperatures (within the range of 12 °C to 30 °C) have extremely different consistencies. The slump of the concrete mixture made at 20 °C (160 mm) was almost twice as high as at 12 °C (280 mm) and almost five times lower than at 30 °C (20 mm). This dependence was demonstrated by keeping the mixture ingredients for 72 h before executing the mixture at the assumed test temperature.

The results of the fresh concrete mixture consistency tests confirm that the HPC mixture is very sensitive to temperature increases. The reason for this could be the faster evaporation of water which, combined with the low water–binder ratio, results in a clear decrease in workability. This effect is also influenced by the acceleration of the cement hydration reaction. Losing workability of a concrete mixture at elevated temperatures can significantly hinder its application. This is indicated by the results of consistency tests as well as air content in the concrete mixture at 30 °C.

Temperature has been shown not to have a significant effect on properties, such as density, water absorption, and depth of water penetration under pressure. The indicated parameters for the concrete samples prepared at different temperatures differed slightly.

Compressive strength tests carried out over a period of 3 days to 2 years on concrete samples prepared at temperatures of 12, 20, 30, and 40 °C showed that the rate of compressive strength growth increases with increases in temperature. The concrete prepared at 40 °C reached 99 MPa after only three days (i.e., 91% of the 28-day strength), while at 12 °C the concrete reached 74.5 MPa (i.e., 70% of the 28-day strength). This confirms that at elevated temperatures the cement hydration rate increases, resulting in a faster increase in concrete compressive strength in the first 28 days of curing.

After 28 days, the highest compressive strength was achieved by the concrete maturing at lower temperatures. At the age of two years, concrete produced at temperatures of 12, 20, 30, and 40 °C reached compressive strengths of 135.6, 134.8, 124.5, and 119.7 MPa, respectively.

As shown, an increase in the temperature of concrete curing results in a decrease in its compressive strength after a long period of time, compared with concrete cured at lower temperatures. This may be due to the fact that a faster rate of hydration results in hydration products with a more irregular structure and higher porosity, which negatively affect compressive strength after a longer curing time.

The results of the freeze–thaw durability tests performed using the Polish standard method and the ‘’slab test’’ show that HPC is resistant to freeze–thaw. The decrease in compressive strength after 300 freezing and thawing cycles was relatively small and amounted to a maximum of 8.6%, compared with samples not subjected to cyclic freezing and thawing, in the case of the concrete prepared at the lowest temperature of 12 °C. For the concrete cured at 20 °C and 30 °C, the strength drop was 4.3% and 1.7%, respectively. This may be due to the fact that, at the start of the test (28th day of curing), the concrete stored at lower temperatures reached a lower compressive strength than the concrete stored at higher temperatures.

The freeze–thaw durability tests in the presence of a NaCl solution showed minimal scaling of the concrete surface, which is proof of very good freeze–thaw durability of the tested HPC, regardless of its preparation temperature.

This study shows a significant influence of lowered and increased temperature on the properties of the concrete mix and hardened HPC concrete, especially in terms of consistency and compressive strength. This points to directions for further research, which should include studies of the rheological parameters of the HPC concrete mix and its changes over time. Research aimed at demonstrating the influence of curing temperature on the cement hydration process for binders used in HPC concrete (with a low water–binder ratio) should also be carried out. Future research should include changes of hydrate phases over time. 

## Figures and Tables

**Figure 1 materials-13-04646-f001:**
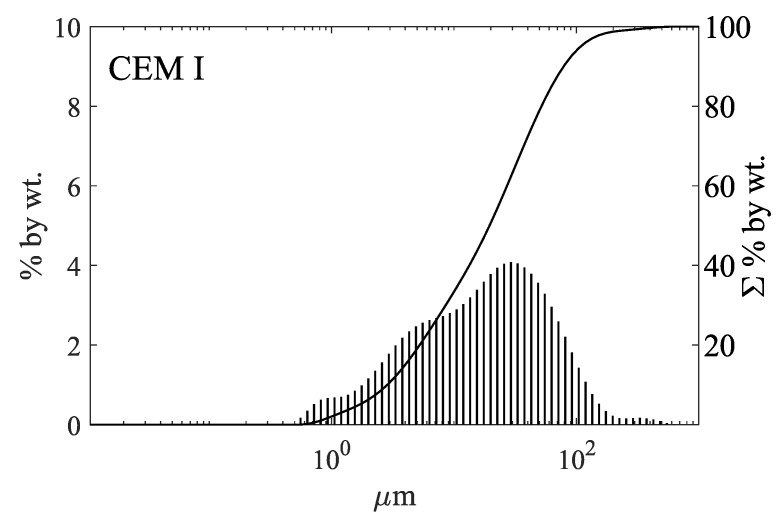
Particle size distribution of Portland cement CEM I 42.5 R (CEM I).

**Figure 2 materials-13-04646-f002:**
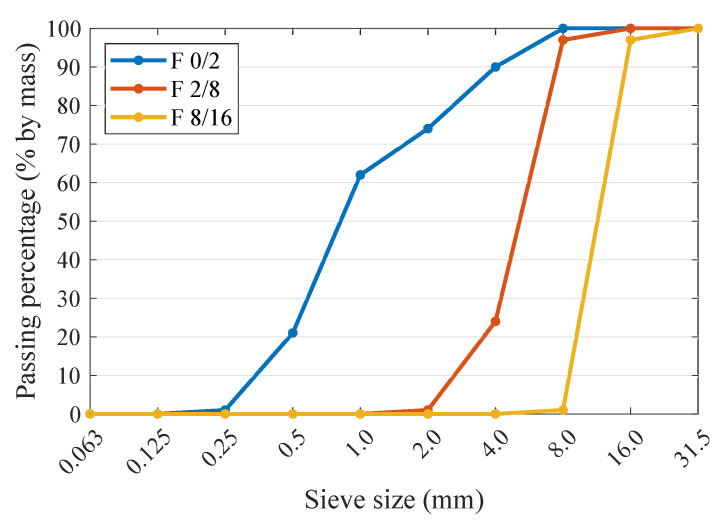
Particle size distribution of aggregates used in the production of HPC.

**Figure 3 materials-13-04646-f003:**
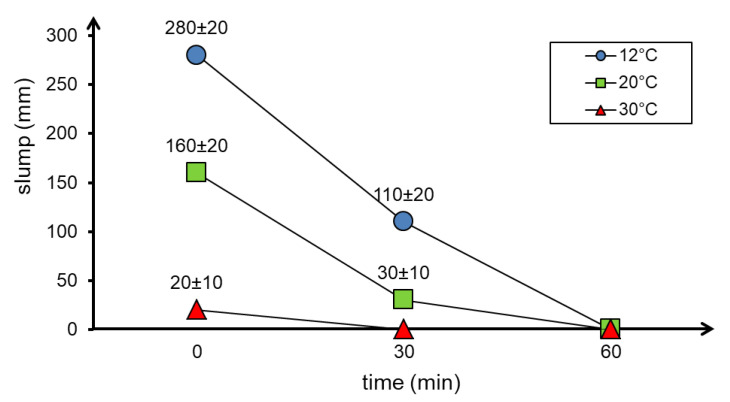
Results of concrete mixture consistency tests with the use of the slump test depending on the temperature and time of testing.

**Figure 4 materials-13-04646-f004:**
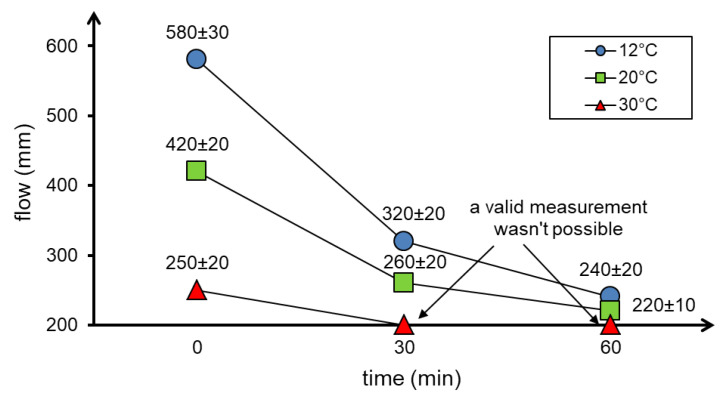
Results of concrete mixture consistency tests with the use of the flow table test depending on the temperature and time of testing.

**Figure 5 materials-13-04646-f005:**
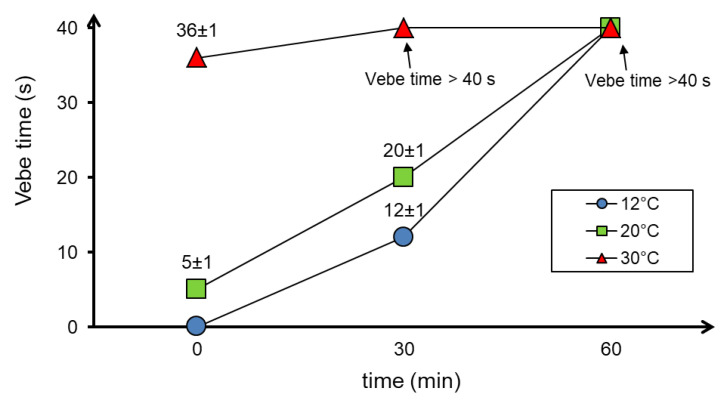
Results of concrete mixture consistency tests with the use of the Vebe test depending on the temperature and time of testing.

**Figure 6 materials-13-04646-f006:**
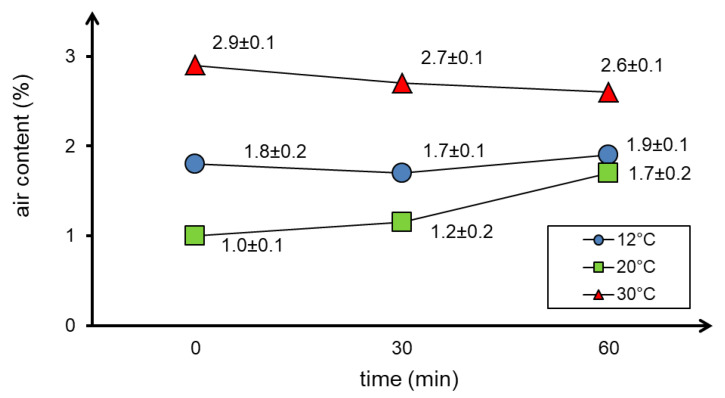
The results of air content tests in the concrete mixture depending on the curing temperature and time.

**Figure 7 materials-13-04646-f007:**
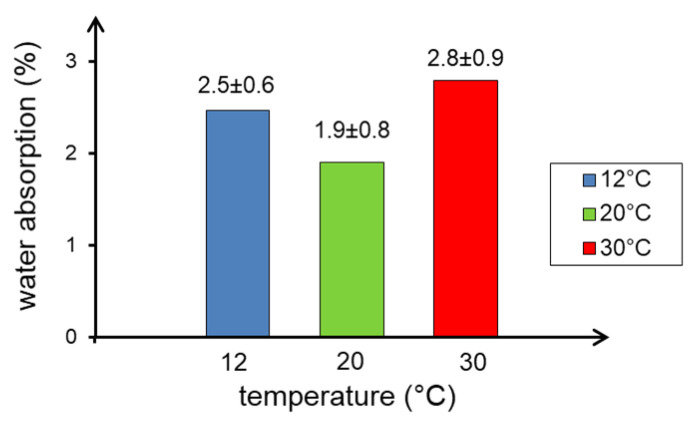
Water absorption of concrete samples depending on concrete curing temperature.

**Figure 8 materials-13-04646-f008:**
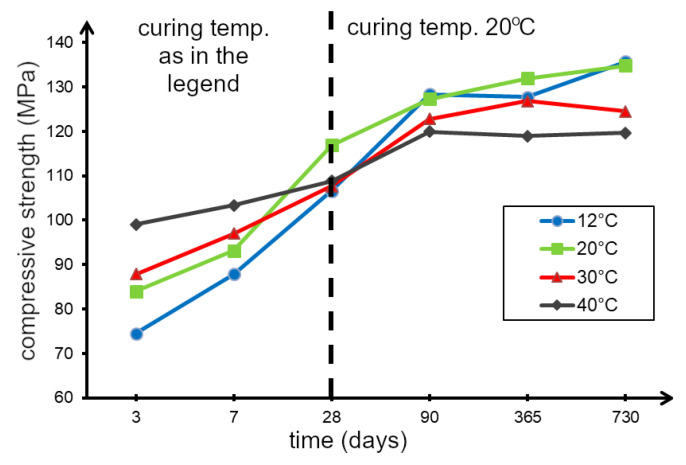
Results of compressive strength tests of concrete cured at 12 °C, 20 °C, 30 °C and 40 °C.

**Figure 9 materials-13-04646-f009:**
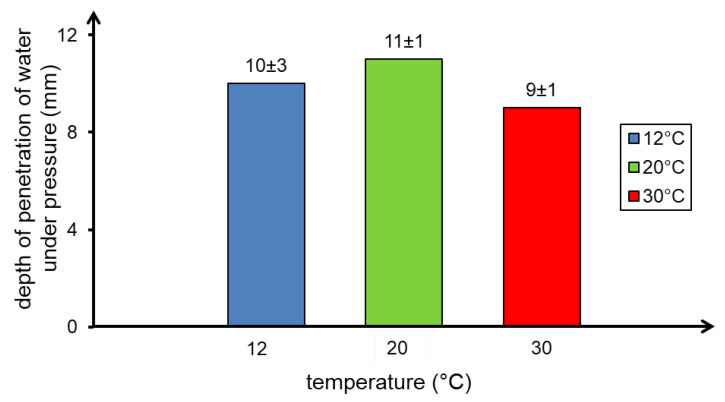
Depth of penetration of water under pressure in concrete samples depending on the curing temperature.

**Table 1 materials-13-04646-t001:** Chemical composition of Portland cement CEM I 42.5 R (% mas.).

Cement	${\mathrm{SiO}}_{2}$	${\mathrm{Al}}_{2}O_{3}$	${\mathrm{Fe}}_{2}O_{3}$	CaO	MgO	Cl${}^{-}$	${\mathrm{Na}}_{2}O_{\mathrm{eq}}$	${\mathrm{SO}}_{3}$	$K_{2}O$
CEM I	21.9	5.8	2.9	63.1	1.2	0.01	0.7	2.1	0.5

**Table 2 materials-13-04646-t002:** Physical properties of aggregates.

Fraction	Average Value		
	Bulk Density	Specific Density	Water Absorption
	(kg/dm${}^{3}$)	(kg/dm${}^{3}$)	(%)
0/2	1.72	2.64	1.4
2/8	1.64	3.12	1.6
8/16	1.60	3.18	0.5

**Table 3 materials-13-04646-t003:** Concrete mixture composition.

Ingredient	Content (kg/m${}^{3}$)
cement	500
0/2 fraction fine aggregate	656
2/8 fraction coarse aggregate	592
8/16 fraction coarse aggregate	740
water	150
silica fume	50
superplasticizer	7.5

**Table 4 materials-13-04646-t004:** Mixing procedure for concrete mixture ingredients.

Time	Activity Performed
(min)	
0–2	mixing the coarse aggregate fractions (2/8 and 8/16 mm)
2–4	adding the fine aggregate fraction (0/2 mm)
4–6	adding the cement and silica fume
6–8	adding 1/2 the water amount
8–14	adding 1/2 the water amount with the superplasticizer
14	mixing completion

**Table 5 materials-13-04646-t005:** Temperature influence on the density of the concrete mixture.

Ambient Temperature	Concrete Mixture Temperature	Compaction Method	Density
(°C)	(°C)		(kg/m${}^{3}$)
12	17.3	rodding/manual	2620 ±10
20	23.5	vibrating table/mechanical	2630 ±10
30	32.7	vibrating table/mechanical	2630 ±10
40	41.6	vibrating table/mechanical	2630 ±10

**Table 6 materials-13-04646-t006:** Influence of ambient temperature on the density of concrete samples.

Curing Temperature	Concrete Density
(°C)	(kg/m${}^{3}$)
12	2650 ± 20
20	2660 ± 10
30	2680 ± 20

**Table 7 materials-13-04646-t007:** Results of compressive strength tests (MPa) of concrete samples cured at 12 °C, 20 °C, 30 °C and 40 °C.

Time (days)	Curing Temperature			
	12 °C	20 °C	30 °C	40 °C
3	74.5 ± 5.3	84.0 ±1.3	87.9 ± 2.5	99.0 ± 9.1
7	87.8 ± 6.1	93.1 ± 4.0	97.0 ± 11.0	103.3 ± 4.6
28	106.6 ± 2.9	116.9 ± 7.6	107.6 ± 6.3	108.8 ± 6.2
90	128.3 ± 10.4	127.3 ± 7.1	122.7 ± 6.9	119.9 ± 10.7
365	127.7 ± 4.8	131.9 ± 1.8	126.8 ± 0.6	119.0 ± 7.0
730	135.6 ± 6.9	134.8 ± 7.9	124.5 ± 3.7	119.7 ± 6.8

**Table 8 materials-13-04646-t008:** Results of the tests of resisting of the concrete samples to cyclic freezing and thawing determined by the method according to PN-B-06250.

Curing Temperature	12 °C	20 °C	30 °C
loss of mass of samples after the test (%)	0.01	0.00	0.03
decrease in strength of samples after the test (%)	8.6	4.3	1.7

**Table 9 materials-13-04646-t009:** Results of the tests of resisting of the concrete samples to cyclic freezing and thawing in the presence of de-icing salt (3% NaCl), cured at 12, 20, and 30 °C.

Curing Temperature	Average Value of the Sample Scaling Mass (kg/m${}^{2}$)				
	After 7 Cycles	After 14 Cycles	After 28 Cycles	After 42 Cycles	After 56 Cycles
12 °C	0.00 ± 0.02	0.00 ± 0.02	0.02 ± 0.02	0.02 ± 0.02	0.02 ± 0.02
20 °C	0.00 ± 0.02	0.02 ± 0.02	0.02 ± 0.02	0.04 ± 0.02	0.04 ± 0.02
30 °C	0.00 ± 0.02	0.02 ± 0.02	0.02 ± 0.02	0.04 ± 0.02	0.04 ± 0.02

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
