# Peer review of "The Influence of Ambient Temperature on High Performance Concrete Properties"

_materials, 2020, doi:10.3390/ma13204646_

Round 1

Reviewer 1 Report

This paper focuses on the influence of ambient temperature on the HPC properties. Author claimed that with the raising temperature the workability of the fresh concrete mixture decreases and the hardened concrete has a higher early strength before 28 days and a lower strength after a period longer than 28 days. Although paper is well-written, the result analyses in this manuscript are inadequate. Major revision should be made as follows:

-Abstract. Why is the highest curing temperature set to 40℃, except for the compressive strength, it is not reflected in other tests. Please determine the assumed temperature for each test and result and revise the Abstract.

-Line 107. “The most susceptible to excessive …………….. significantly affects the properties of the hardened concrete” references should be added. Please give the conclusion of their study if it is available.

-Line 131. “The tests were carried out at both increased (30℃) and lowered (12℃) concrete curing temperature”. At the following tests, the highest temperature is 40℃.

-Line 264. “The concrete mixture density tests showed that the temperature in the tested range (from 12℃ to 30℃)” is different from the results given in the Tab.5 (from 12℃ to 40℃).

-Fig. 8. Why does the strength of hardened concrete with 12℃ decrease at 365d.

-Line 283. “However, after a longer curing period (from 28 days to 2 years), the  highest strength was demonstrated for concrete cured at lower temperatures.” may be wrong. The data presented in Fig.8 show that the strength(20℃>40℃>30℃>10℃) of the hardened concrete is not higher with the lower temperature at the 28d. The time should be set at an accurate value. The Fig.8 needs further analysis.

-Line 285. “Fig. 8 show that for concrete” should be changed into “Fig. 8 shows that for concrete”.

-Line 288. “can be explained by the acceleration of the cement hydration process [37]. However, as the results of the research [25] show”. The compressive strength of hardened concrete in the initial period of time(up to 7 days) is different from a longer curing period (over 90 days). Authors should give the experimental evidence about the influence of curing temperature on the compressive strength of HPC. For example, the changes of the hydrate phases or C−S−H phase content by in-site XRD analysis.

-Line 328. “the slump was almost twice as high as at 12℃ and almost five times lower than at 30℃”. Please give an accurate value for the result of analysis.

-Line 337. The conclusion “Temperature has been shown not to have a significant effect on properties such as density, water absorption and depth of water penetration under pressure” does not match “The results of the water absorption tests of the samples of concrete cured at different temperatures showed that the least absorbent is the concrete cured at 20℃, and the largest is the concrete cured at 30℃ (Fig. 7)”.

Author Response

Response to Reviewer 1 Comments

This paper focuses on the influence of ambient temperature on the HPC properties. Author claimed that with the raising temperature the workability of the fresh concrete mixture decreases and the hardened concrete has a higher early strength before 28 days and a lower strength after a period longer than 28 days. Although paper is well-written, the result analyses in this manuscript are inadequate. Major revision should be made as follows:

Point 1: Abstract. Why is the highest curing temperature set to 40℃, except for the compressive strength, it is not reflected in other tests. Please determine the assumed temperature for each test and result and revise the Abstract.

Response 1: We agree with the reviewer, the description may mislead the reader. The set of tests was carried out for 12°C, 20°C and 30°C. The following sentence has been introduced into the abstract:

This paper presents the results of tests on High Performance Concrete (HPC) prepared and cured at various ambient temperatures, ranging from 12°C to 30°C (the compressive strength and concrete mix density was tested also at 40°C)

Point 2: Line 107. “The most susceptible to excessive …………….. significantly affects the properties of the hardened concrete” references should be added. Please give the conclusion of their study if it is available.

Response 2: The reference [29] has been added. The text of the paragraph has been modified:

The most susceptible to excessive heating are massive elements, where the cooling surface is small in relation to the mass of the concrete mixture being casted. The negative phenomena caused by excessive heating can be minimized by proper selection of binder composition [29].

Point 3: Line 131. “The tests were carried out at both increased (30℃) and lowered (12℃) concrete curing temperature”. At the following tests, the highest temperature is 40℃.

Response 3: The set of tests was carried out for 12°C, 20°C and 30°C. Additionally the compressive strength and concrete mix density tests were carried out at 40°C. The clarifying sentence was added:

… but in range of practical applicability of concrete. The compressive strength and concrete mix density were tested at temperatures of 12°C, 20°C, 30°C and 40°C. Special attention was paid to achieving the desired temperature ...

Point 4: Line 264. “The concrete mixture density tests showed that the temperature in the tested range (from 12℃ to 30℃)” is different from the results given in the Tab.5 (from 12℃ to 40℃).

Response 4: Thank you for this remark, the text has been corrected:

“The concrete mixture density tests showed that the temperature in the tested range (from 12℃ to 40℃) …” 

Point 5: Fig. 8. Why does the strength of hardened concrete with 12℃ decrease at 365d.

Response 5: The actual research results are given in the paper. It is obvious that the value of the compressive strength should not decrease as it happens in the indicated place. However, this decrease is 0.6 MPa, which is a small value in relation to the estimated measurement uncertainty (given in Table 7).

Point 6: Line 283. “However, after a longer curing period (from 28 days to 2 years), the  highest strength was demonstrated for concrete cured at lower temperatures.” may be wrong. The data presented in Fig.8 show that the strength(20℃>40℃>30℃>10℃) of the hardened concrete is not higher with the lower temperature at the 28d. The time should be set at an accurate value. The Fig.8 needs further analysis.

Response 6: Thank you for this remark. The authors meant "after 28 days". The text has been corrected:

However, after a longer curing period (from 90 days to 2 years), the highest strength was demonstrated for concrete cured at lower temperatures.

The further analysis was presented in the paragraph (lines 295-299):

Concrete cured at 12°C and 20°C achieved similar strength values at the age of 2 years (about 130 MPa), while concrete made at 20°C achieved this strength much faster. It can therefore be concluded that a temperature of 20°C is the most favourable, due to the increase in strength over time. ecreased early concrete strength at low temperatures is also confirmed by the results of the research presented e.g. in the paper [22].

Point 7: Line 285. “Fig. 8 show that for concrete” should be changed into “Fig. 8 shows that for concrete”.

Response 7: Thank you for this remark, the text has been corrected.

Point 8: Line 288. “can be explained by the acceleration of the cement hydration process [37]. However, as the results of the research [25] show”. The compressive strength of hardened concrete in the initial period of time(up to 7 days) is different from a longer curing period (over 90 days). Authors should give the experimental evidence about the influence of curing temperature on the compressive strength of HPC. For example, the changes of the hydrate phases or C−S−H phase content by in-site XRD analysis.

Response 8: In the present study, no analysis of cement hydration products was performed. The results of the obtained tests were only compared with the results of physicochemical tests by other authors (e.g. [37]). The analysis of HPC cement hydration products at different temperatures should be the subject of further research.

Point 9: Line 328. “the slump was almost twice as high as at 12℃ and almost five times lower than at 30℃”. Please give an accurate value for the result of analysis.

Response 9: Thank you for this remark, the accurate value of test results has been given:

… . The slump of concrete mixture made at 20°C (160 mm) was almost twice as high as at 12°C (280 mm) and almost five times lower than at 30°C (20 mm). …

Point 10: Line 337. The conclusion “Temperature has been shown not to have a significant effect on properties such as density, water absorption and depth of water penetration under pressure” does not match “The results of the water absorption tests of the samples of concrete cured at different temperatures showed that the least absorbent is the concrete cured at 20℃, and the largest is the concrete cured at 30℃ (Fig. 7)”.

Response 10: Thank you for this remark. The results of the water absorption test vary slightly depending on the curing temperature but it should be noted, that in all investigated cases the obtained values are at a very low level (comparing to the water absorption of normal concrete). The clarifying sentence has been introduced:

The results of the water absorption tests of the samples of concrete cured at different temperatures showed that in all cases the obtained values were at a very low level. The least absorbent is the concrete cured at 20℃, and the largest is the concrete cured at 30℃ (Fig. 7), however the presented differences should be considered as small. …

Reviewer 2 Report

The research paper investigates the properties of High Performance Concrete (HPC) at various temperatures at 12°C to 40°C. The properties were analyzed by dividing them into fresh concrete mixture and hardened concrete, and in the former, consistency, air content, and density were studied. In the latter case, density, water absorption, depth of water penetration under pressure, compressive strength and freeze-thaw durability of hardened concrete were studied. This provides guidelines for ambient temperature suitable for prepare and cure of HPC. However, the reviewer has to point out several issues to be seriously taken into account before publication.

  1. The authors must clarify the background knowledge to support the significance of the study.
  2. The author cited a case of prior research and mentioned that the hardened cement paste at 60°C has a strong compressive strength due to porosity. This differs from the author's comments regarding temperature increases, porosity and strength. The relationship between curing period, temperature, porosity and strength must be clarified.
  3. Insert the label in Figure
  4. Please provide with the limits, prospects, directions that you haven't dealt with.

Author Response

Response to Reviewer 2 Comments

The research paper investigates the properties of High Performance Concrete (HPC) at various temperatures at 12°C to 40°C. The properties were analyzed by dividing them into fresh concrete mixture and hardened concrete, and in the former, consistency, air content, and density were studied. In the latter case, density, water absorption, depth of water penetration under pressure, compressive strength and freeze-thaw durability of hardened concrete were studied. This provides guidelines for ambient temperature suitable for prepare and cure of HPC. However, the reviewer has to point out several issues to be seriously taken into account before publication.

Point 1: The authors must clarify the background knowledge to support the significance of the study.

Response 1: In the introduction, the works related to the study of the influence of temperature on the properties of cement pastes, a concrete mix and hardened concrete have been widely described. Justification for undertaking HPC concrete tests in this area is presented at the end of the introduction, i.e. in lines 117-129.

Point 2: The author cited a case of prior research and mentioned that the hardened cement paste at 60°C has a strong compressive strength due to porosity. This differs from the author's comments regarding temperature increases, porosity and strength. The relationship between curing period, temperature, porosity and strength must be clarified.

Response 2: The papers cited in the literature survey indicate that binders cured at elevated temperature are characterized by greater porosity (after longer times), which negatively affects the compressive strength (for example lines:38-43, 51-55). This trend is consistent with the results obtained in this study.

In this study, the porosity of the cement matrix maturing at various temperatures was not investigated. The presented comments in this regard are based on the results of research by other authors, e.g. [7-11].

Point 3: Insert the label in Figure

Response 3: Thank you for this remark, the labels have been added.

Point 4: Please provide with the limits, prospects, directions that you haven't dealt with.

Response 4: Thank you for this remark the text has been added:

This study shows a significant influence of lowered and increased temperature on the properties of the concrete mix and hardened HPC concrete, especially in terms of consistency and compressive strength. This points to directions for further research, which should include studies of the rheological parameters of the HPC concrete mix and its changes over time. Research aimed at demonstrating the influence of curing temperature on the cement hydration process for binders used in HPC concrete (with a low water–binder ratio) should also be carried out. Future research should include changes of hydrate phases over time.   

Reviewer 3 Report

The English is not really poor but it is recommended that an English speaking person corrects the text. There are numerous errors and strange expressions such as line 217: 4 cubic cubes. It would be more informative to show the time-dependent amount of absorbed water instead of Fig. 7. In the conclusions it is stated that the influence of temperature on properties of concrete is important due to climate change. Climate change is generally estimated to remain below 2 °C! Line 340: ... concrete samples prepared at temperatures of 12 °C, 20 °C, 30 °C, and 40 °C showed that the rate of compressive strength growth over time increases with temperature. In the list of references many papers such as Ref 3 are abreviated: B. Lothenbach et al. all authors should be indicated. 

Author Response

Response to Reviewer 3 Comments

Point 1: The English is not really poor but it is recommended that an English speaking person corrects the text. There are numerous errors and strange expressions such as line 217: 4 cubic cubes.

Response 1: The proof-reading correction has been done. Thank you for this remark, the errors has been corrected as far as possible.

Point 2: It would be more informative to show the time-dependent amount of absorbed water instead of Fig. 7.

Response 2: We agree with the reviewer's suggestion. Unfortunately, the water absorption tests were carried out in accordance with the procedure (described in chapter 3.3), which involves soaking of samples that were not completely dried before. The results which we can present would therefore be incomplete, therefore we decided not to implement the suggested correction.

Point 3: In the conclusions it is stated that the influence of temperature on properties of concrete is important due to climate change. Climate change is generally estimated to remain below 2 °C!

Response 3: We agree with the reviewer’s comment that the importance of climate change is debatable. The sentence has been changed:

HPC is increasingly used in civil engineering. The assessment of the influence of temperature on the properties of concrete is particularly important because of the applying HPC concrete at both elevated and lowered temperatures. The paper presents the results of research on the influence of ambient and curing temperature on the properties of the concrete mixture and hardened HPC.

Point 4: Line 340: ... concrete samples prepared at temperatures of 12 °C, 20 °C, 30 °C, and 40 °C showed that the rate of compressive strength growth over time increases with temperature.

Response 4: Thank you for this remark the text has been changed:

… concrete samples prepared at temperatures of 12°C, 20°C, 30°C, and 40°C showed that the rate of compressive strength growth increases with increase in temperature.

Point 5: In the list of references many papers such as Ref 3 are abreviated: B. Lothenbach et al. all authors should be indicated. 

Response 5: Thank you for this remark the references has been corrected.

Round 2

Reviewer 3 Report

This paper has substantially improved.